# Health and Socioeconomic Determinants of Abuse among Women with Disabilities

**DOI:** 10.3390/ijerph20126191

**Published:** 2023-06-20

**Authors:** Javier Zamora Arenas, Ana Millán Jiménez, Marcos Bote

**Affiliations:** 1Attention to Diversity and Volunteering Service, University of Murcia, 30100 Murcia, Spain; javierzamora@um.es; 2Department of Sociology, Faculty of Economics and Business, University of Murcia, 30100 Murcia, Spain; amillan@um.es

**Keywords:** violence, gender, disability, women, vulnerability, social risk

## Abstract

The double vulnerability of women with disabilities places them at the center of this research paper. Intersectionality is key in research on gender-based violence. This study analyzes the perspective of the victims and non-victims themselves on this issue, through a comparative analysis between women with and without disabilities, at two levels of analysis: quantitative, through the adaptation of various scales (Assessment Screen-Disability/AAS-D, and the Woman Abuse Screening Tool/WAST), and qualitative, with semi-structured interviews (open scripts and different themes), and focus groups with experts from the associative network. The results obtained indicate that the most frequent type of violence is physical, followed by psychological and sexual, mainly perpetrated by partners. The higher their level of education, the more they defend themselves; receiving public aid can be a risk factor for domestic and sexual violence, and belonging to the associative movement and having paid work outside the home act as preventive measures. In conclusion, it is necessary to establish strategic protection measures and effective detection and intervention systems to make victims visible and care for them.

## 1. Introduction

Disability is an issue that currently occupies and concerns society in general, and institutions in particular. As societies have achieved a higher degree of development and improved the quality of life of their citizens, they have expanded their focus of attention to previously considered secondary issues, such as disability. It is true that the understanding of disability today is a relatively recent concept, as it has evolved over time, like all words. Even now, there are several interpretations and models that attempt to explain it [1].

A brief historical overview of the term reveals its connection to different cultural elements [2], which means it has undergone its own modifications. These physical, psychological, or intellectual characteristics that are defined as impairments have been associated with various myths and beliefs, eliciting both supportive and socially rejecting responses. It was in the 19th century when scientific and rational explanations began to emerge, and governments started paying attention to people with disabilities. This governmental shift became a turning point in favor of their social inclusion.

We must fast forward to the second half of the 20th century, with the rise of the welfare state and the renewal of the concept of citizenship, for preventive and inclusive measures to be implemented. Special centers for work and education, as well as associations and foundations for people with disabilities and their families, emerged. It was then that disability gained importance and presence, both socially and academically.

As a result of this emerging interest, theoretical models, even antagonistic ones, emerged to conceptualize disability [3]. There are four paradigms that have had the most impact and influence on the current social landscape: the Individual Model, the Social Model, the Biopsychosocial Model, and the Diversity Model. It is important to emphasize that these models of conceptualization and interpretation generate social representations and perspectives that form the basis of current legislation, state and institutional social policies and interventions, initiatives promoted by non-governmental organizations and private companies, professional practices and services, academic studies and research, and the attitudes of societies in general [4]. In other words, the way disability is conceived has direct and significant implications for the individuals involved and their environments. The Individual Model understands disability in terms of the consequences it entails. Consequences must be corrected (as much as possible), and the ultimate goal is to achieve the maximum possible standardization.

The Social Model argues that the conception of disability is a socially constructed imposition and a form of social oppression resulting from a society that does not consider the collective of people with disabilities. It advocates for autonomy in deciding one’s own life and the elimination of any barriers that hinder equal opportunities [5]. It supports inclusion and values that promote independent living, non-discrimination, universal accessibility, and normalization of the environment. The goal is to minimize any type of limiting barrier, whether physical, social, procedural, or behavioral. As Verdugo points out, the Social Model is fundamentally a theoretical development that arises as a result of struggles for independent living, citizenship, or civil rights for people with disabilities [6,7].

The third model is the Biopsychosocial Model, proposed by the World Health Organization (WHO), which sees itself as an integrative framework for the previous two models, recognizing that neither of them alone can fully encompass and explain the complex reality of disability. It is important to connect the biological and social aspects [8,9]. The WHO responded to this by publishing the International Classification of Functioning, Disability, and Health (ICF) in 2001, which improved upon previous versions (ICIDH) and aimed to provide a new synthesizing and descriptive model of health and the environmental factors related to it [10].

Actually, a disability is much more than a lack of something, and to fully understand it, it is necessary to distinguish between impairments, disabilities, and handicaps, and emphasize the implications they entail. Environmental factors can positively or negatively influence the functioning of people with disabilities. Their absence or presence can either limit or facilitate their activities. The result of these considerations is the expanded concept and definition of disability as a general term encompassing impairments, activity limitations, and participation restrictions. Impairments refer to problems that affect a person’s body structure or function; activity limitations are difficulties in executing actions or tasks, and participation restrictions are problems in engaging in life situations.

Lastly, the Diversity Model emerged through the work of Javier Romañach and Agustina Palacios [11,12], inspired by the Independent Living Movement (ILM). They advocate for recognizing the intrinsic diversity in every human being and propose the integration of bioethics and the foundations of Human Rights as tools to achieve a deep, necessary, and essential social change toward full equality of opportunities and the non-discrimination of people with disabilities. They adopt the term ‘functional diversity’ and reject words such as ‘handicap’ and ‘disability’ due to their negative connotations [12]. The central idea is that diversity and plurality enrich humanity. The unique bodies and abilities of individuals should not be despised or discriminated against. There are people with different bodies and abilities that function in multiple ways, different from most of the population. Thus, the concept of ‘normal’ is merely quantifiable, referring to what most people do or how they do it, but it is only a reference to what is most common.

In another matter, according to the 2017 report of the United Nations Special Rapporteur on the rights of persons with disabilities, women, young girls, and girls with disabilities are disproportionately affected by various forms of gender-based violence. These forms of violence include physical, psychological, and emotional abuse, sexual abuse, harassment, coercion, arbitrary deprivation of liberty, confinement, infanticide of girls, human trafficking, prostitution, neglect, domestic violence, and harmful practices such as child and forced marriage, female genital mutilation, forced sterilization, and invasive and irreversible treatments [13]. As pointed out by Elena Díaz [14], women and girls with disabilities are more exposed to violence, abuse, and mistreatment. They also have less access to culture, education, and healthcare, which makes them highly vulnerable. It is a multiple and limiting discrimination that hinders their ability to exercise their sexual, reproductive, social, educational, labor, and healthcare rights.

In conclusion, it is necessary to delve deeper into the issue of women with disabilities in today’s complex society. These individuals face multiple disadvantages, namely, disability and gender, along with variables such as social class, income level, age, ethnicity or race, and habitat. Pineda and Luna [15] argue that there is no primacy of the condition of disability or being a woman, but rather an explicit reality of both conditions that we perceive discursively through fragmented categories of gender and disability, resulting in a unique form of discrimination that could be called ‘gendisability’, turning them into perfect targets.

If we look at recent data provided by the United Nations (2021), the average prevalence of women with disabilities (over 18 years old) is 19.2%, whereas the average prevalence of men with disabilities is 12%, which means that approximately one in five women has some type of disability. These data justify the need to analyze the situation of these women from a cross-cutting perspective. The Istanbul Convention (Council of Europe Convention on preventing and combating violence against women and domestic violence, 11 May 2011, signed and ratified by the Spanish government) indicates that it is essential to ‘recognize that the structural nature of violence against women is based on gender, and that this violence is one of the crucial social mechanisms by which women are kept in a subordinate position to men’. In other words, gender-based violence is the most extreme manifestation of the structural and historical inequality between women and men, and it is one of the most alarming social scourges our society faces.

In 2012, the Office of the United Nations High Commissioner for Human Rights estimated that nearly 80% of women with disabilities are Victims of Violence and have a four times higher risk of experiencing sexual violence than other women. It stated that 80% of women with disabilities living in institutions suffer violence from people in their environment. Girls with disabilities are at a higher risk of experiencing violence and harmful practices such as infanticide, early and forced marriage, forced sterilization, female genital mutilation, rape of virgins, exploitation, and human trafficking [16]. This situation of helplessness, dependence, vulnerability, or lack of support easily leads to situations of violence. As indicated by Verdugo and colleagues [17], most research on sexual abuse and intellectual disabilities identifies attitudes, beliefs, myths, and false prejudices about the sexuality of this group as triggering factors. Specifically, it seems that the vulnerability of women with disabilities concerning violence is linked to two key factors: the belief that men and women are not equal (resulting in social and physical domination) and the social perception of disability, which views the bodies of disabled individuals as different from the rest, incapable, sick, and therefore ‘abnormal’.

Another example of abusive behavior is related to healthcare and reproductive rights. In this regard, women with disabilities start from a disadvantaged position and also have less access to existing resources for family planning, both due to existing prejudices and lack of information and accessibility [14,18]. That is why the CERMI Women Foundation, in 2021, in its Protocol for the Care of Women with Disabilities Who Are Victims of Violence, states that women with disabilities are often invisible in the healthcare system and in action plans addressing violence [19].

In conclusion, the data are overwhelming and highlight the need to improve the quantity and quality of preventive and support resources, which should be focused on considering women as rights holders.

## 2. Research Aims

The main objective of this article is to understand the reality of women with disabilities who are victims of gender-based violence at a regional level. It aims to identify the specific variables underlying situations of perceived and experienced violence, positioning and assessing them within the general framework provided by the Spanish State. The specific objectives are as follows:

1. Determine the biopsychosocial variables of women with disabilities. Specifically, origin and location, age, personal economic context, family and familial environment, support network, educational context, personal work and economic context, and awareness of available resources.

2. Understand and define the situations of gender-based violence experienced by women with disabilities and their physical, psychological, and social impact (characteristics of the violence, types of perceived and experienced violence, duration, background, and consequences on physical, psychological, and social levels).

3. Conduct a comparative analysis of the obtained data: compare data from victims of different types of violence (based on age, biopsychosocial situation, type, and degree of disability) by conducting all relevant variable crosses for the study and compare the results with general data on violence against women without disabilities.

This research has been conducted with funding from the Ministry of Women, Equity, LGBTI, Family, and Social Policy of the regional government of Murcia (Spain) with the purpose of studying violence against women with disabilities. Consequently, men have not been included in this sample. Transgender women were not excluded from recruitment; however, no transgender women participated in the study.

## 3. Data and Methods

For this research, several methodologies (quantitative and qualitative) have been used, depending on the themes, type of content being investigated, and the desired social intervention. A cross-sectional, non-probabilistic, and intentional design was employed, as the sample consists of women diagnosed with disabilities, encompassing various types and degrees.

The data were obtained first through the administration of a questionnaire which was distributed both online (71 women) and administered by a team of 5 interviewers (110 women) who conducted face-to-face interviews. A total of 181 women with disabilities from 14 different organizations (ASTUS-PROLAM, ONCE, FUNDOWN, AFEMAC, PROMETEO, AFEMCE, AFEMAR, ASOFEM, APCOM, AFEMTO, AIDEMAR, AFESMO, AFES, and AFEMNOR) located in 11 different municipalities in the Region of Murcia comprised the final sample.

This questionnaire, which was also used as a reference for the semi-structured interview, utilized instruments commonly used in this type of research. These instruments include the Assessment Screen-Disability (AAS-D) provided by McFarlane, Hughes, Nosek, Groff, Swedlend, and Dolen Mullen [20], specific violence-related questions for women with disabilities from the Social Support Inventory (AS) provided by Matud, Padilla, and Gutiérrez [21], the Macro-Survey on Violence against Women [22], the Woman Abuse Screening Tool (WAST) provided by Fogarty and Brown [23], the EISS (Social Integration and Health Survey) provided by the National Institute of Statistics [24], and, especially, the Protocol for the Attention to Women with Disabilities Who Are Victims of Violence [19].

In the second phase, personal testimonies were obtained from 18 women with different disabilities who had already responded to the surveys and expressed their willingness to participate in personal interviews. Simultaneously, a focus group was conducted with the participation of 7 women representing disability associations (Militares y Guardias Civiles con Discapacidad Murcia (ACIME), FUNDOWN, FAMDIF, and Más Mujer association, Plena Inclusión Región de Murcia, Delegación Territorial de la ONCE Región de Murcia, and CERMI Región de Murcia). Table 1 contains a summary of the profile of organizations.

Interviews: Taking the questionnaire as a reference, qualitative information is obtained regarding situations of gender-based violence experienced by women with disabilities in the Region of Murcia. Discourse analysis is employed, following the model by Wetherell and Potter [25]. Subsequently, the results and conclusions reached in this phase of the research are compared with those obtained after administering the initial questionnaire and are presented in the final part of this article. The interviews, prepared to be conducted individually, took place in different municipalities (Murcia, Cartagena, Molina de Segura, Cieza, Águilas, Torre Pacheco, Cehegín, San Javier, San Pedro del Pinatar, and Las Torres de Cotillas) in the Region of Murcia, where the participants resided or carried out their daily activities. These were semi-structured interviews with open-ended scripts, organized into thematic topics consisting of 40 categories or codes, distributed across 15 secondary sections that form the 3 main sections, as presented in Table 2, which capture data related to gender-based violence experienced and perceived by the analyzed group.

Focus group: a group of 7 experts from various associations forming the CERMI women’s platform, who are also part of the ÚNICAS project (a program aimed at conducting workshops in different municipalities to work on the empowerment of women with disabilities, identify problematic situations, and plan and implement positive action plans for women).

Techniques such as reading, understanding, and summarizing are employed in the selection of the consulted information. The evaluation criteria for sources include comprehensiveness, logical coherence, and appropriate presentation. The analysis of the sample is conducted using basic frequency analysis for qualitative variables and measures of central tendency for quantitative variables. The IBM SPSS 24 statistical package for Macintosh is used for data analysis, and discourse analysis procedures from the theories of Strauss and Corbin [26] are employed. The analysis of discourse content allows for the acquisition, in an inferential manner, of meaning schemes that can explain one or several dimensions of reality, a state, and a context. Due to the active dimension of the language used by the participants and the conceptual interconnectedness of this methodology, it reinforces and serves as support, through these testimonies, to expand and redefine the information obtained in a systematic way through quantitative techniques. This is particularly relevant in a matter such as gender-based violence towards women with disabilities, characterized by the intersectionality of its defining elements.

### Sampling

The initial sample consists of 181 women, and those who have been assaulted by their current or former partners are selected (158).

In the qualitative analysis, individual interviews are conducted with 18 out of the 158 women. The age range is between 21 and 74 years, with a mean age of 42.72 years (SD = 14.41). Regarding the type of disability, the represented women have a diagnosis of psychosocial disability (*n* = 8), intellectual disability (*n* = 4), physical disability (*n* = 1), organic disability (*n* = 1), mobility problems (*n* = 2), and multiple diagnoses (psychosocial disability as the main diagnosis and other secondary diagnoses) (*n* = 2). Table 2 summarizes this information.

The relevant sociodemographic information for the study is distributed across 5 age intervals: from 21 to 30 years, 17.1%; from 31 to 40 years, 25.3%; from 41 to 50 years, 22.2%; from 51 to 60 years, 22.8%; and 61 years and above, 10.8%. There is a loss of 1.9% of data where participants did not indicate their age. The median age is 45 years, and the mean age is 44.21 years (SD: 13.35). Regarding their place of residence (municipality at the time of the survey), the majority of women reside in urban areas. Out of the total, 116 (73.4%) live with family members, 22 (13.9%) live alone, and the remaining live in shared apartments, residential facilities, or other options.

The recognized degree of disability is organized into three intervals: 33 to 64% (37.3%), 65 to 75% (33.5%), and over 75% (12.0%). The median is 65% (SD: 16.16).

Furthermore, the educational level of the sample is highly heterogeneous, so it is decided to group them into two categories: higher education (university and higher degree, 17.7%) and non-higher education (80.4%).

The type of recognized disability is distributed as follows: intellectual (57.6% of participants) and non-intellectual (42.4%). Finally, regarding membership in the associative movement, it should be noted that most of them (51.3%) belong to some organization.

Regarding the employment status of the respondents, 32.3% are employed, while 67.7% are unemployed. Table 3 contains a summary of all this information.

## 4. Findings

The results are presented by differentiating the types of data obtained from both quantitative and qualitative analyses.

### 4.1. Quantitative Analyses

The non-parametric analysis is conducted using tests and measures of association for two-dimensional tables.

#### 4.1.1. Experienced Abuse: Having Experienced Any Type of Abuse from Their Current or Former Partners

Initially, a frequency analysis is conducted on the responses regarding whether they have been victims of any form of violence, including physical, sexual, psychological, emotional, or instrumental abuse from their partners or former partners.

Data show that only 41.1% affirm having experienced abuse. This percentage contrasts with the results obtained when asking about specific types of violence, indicating that some of them may not be aware of being subjected to abuse (Figure 1).

##### Education Level

The cross-tabulation of the variables ‘experienced abuse’ and ‘education level’ (higher education and non-higher education) shows that having a better education or a university degree is a protective factor against abuse. Women with disabilities who have a higher level of education are less likely to experience abuse compared to women with little or no education (χ^2^ = 3.86; α = 0.049). Please refer to Table 4 for more details.

##### Received Economic Assistance

The cross-tabulation of the variables ‘financial assistance’ and ‘experienced abuse’ reveals that women with disabilities who do not receive any form of economic assistance are less likely to experience abuse compared to those who receive such assistance (χ^2^ = 8.24; α = 0.041). In other words, being a beneficiary of public assistance becomes a risk factor for experiencing abuse. Please refer to Table 4 for further details.

In the non-parametric analyses between the variable ‘abuse’ and other factors such as age, type and degree of disability, employment status, and affiliation with disability organizations, it is concluded that there are no statistically significant differences to explain the perceived and experienced abuse situations. Please note that further details can be found in the respective statistical analyses, but the results suggest that these variables may not play a significant role in explaining the occurrence of abuse.

#### 4.1.2. Physical Abuse: Having Suffered Physical Abuse at the Hands of Their Partners or Ex-Partners

The frequency analysis for the variable ‘physical abuse’ reveals that 30.4% of women confirm having experienced physical aggression, while 69% report not experiencing it (Table 5 and Figure 2). Therefore, physical abuse is not the most common form of gender-based violence against these women. There are other more prevalent forms of abuse and aggression perpetrated by partners and ex-partners.

##### Education Level

Once again, it is demonstrated that having a higher level of education protects against this kind of violence (χ^2^ = 4.25; α = 0.039). Please refer to Table 5 for more details.

##### Membership in the Associative Movement

In the same way, as in the previous case, it is confirmed again that belonging to the associative movement acts as a protective factor against physical abuse (χ^2^ = 6.20; α = 0.045) (Table 5).

##### Working Status

Women who work (remunerated work outside the domestic sphere) experience less physical abuse compared to those who do not work (χ^2^ = 4.28; α = 0.039) (Table 5).

The non-parametric analyses in conjunction with other variables indicate that there are no statistically significant differences, sufficient to explain the situations of perceived and received physical abuse. However, without becoming a statistically significant difference, the analysis between this type of abuse and the recognized disability (intellectual and non-intellectual) indicates a slight tendency for women with an intellectual disability to suffer more physical abuse in comparison with women with non-intellectual disability.

#### 4.1.3. Sexual Abuse: Having Experienced Situations of Sexual Abuse by Their Partners or Ex-Partners

According to the frequency analysis conducted for the variable ‘sexual abuse’, it is found that 87.3% of respondents have not been victims of this type of abuse, while 12.0% have experienced it (Figure 3).

##### Working Status

Women who have salaried employment experience less sexual abuse compared to those who do not have it (χ^2^ = 4.75; α = 0.029). Table 6 contains a summary of all this information.

As was the case before, women who receive financial assistance have experienced more sexual abuse from their spouses or ex-spouses in comparison to those who do not receive such assistance (non-statistically significant) (Table 7).

#### 4.1.4. Other Types of Gender-Based Violence: Having Been Victims of Other Forms of Mistreatment or Aggression (Different from the Previous Ones) Perpetrated by Their Partners or Ex-Partners

In this section, other types of mistreatments executed by the partners or ex-partners of women with disabilities are analyzed (insults, contempt, threats, control, and humiliation). Notably, 50.6% of women declare not experiencing this mistreatment, while the remaining 48.7% confirm that they do (Figure 4).

Despite these similar percentages, it continues to be evident (non-statistically significant) that women who are beneficiaries of economic assistance are more likely to experience mistreatment, especially when the assistance is received and managed by another person, in the light of the data obtained (χ^2^ = 7.56; α = 0.056).

### 4.2. Qualitative Analyses

#### 4.2.1. Face-to-Face Interviews with Women with Disabilities

From the content analysis, it is concluded that, in general, women with disabilities who have experienced abuse, often by their former partners, find it difficult to initiate social relationships and are particularly reluctant to enter new romantic relationships. Many of them were coerced into sexual relationships, especially women with a diagnosis of intellectual disability, and this abuse intensified over time. That implies that almost all the interviewed women (regardless of age, background, or diagnosis) have indicated that they have felt (and continue to feel, at present) fear, insecurity, and social exclusion, derived from and related to the situations of violence they have experienced, justified by the memories and the fear that the episodes will be repeated (with previous perpetrators or potential new aggressors), as well as distrust towards the individuals who carried out the violence, since many of them are forced to interact with their abusers for economic and/or administrative reasons. In this way, women with intellectual disabilities and psychosocial disabilities are the ones who work the least and have the least economic control. They also do not have easy access to their own belongings, nor do they always have what they need or want when they want it, although they may occasionally manage to obtain it.

No, no, it’s me, it’s been since I was 20 when I’ve had any relationship and they’ve told me they wanted to have children… After being abused I’ve suffered, and I don’t want another person to suffer in life because of me. I don’t feel prepared, I don’t feel responsible enough to take care of someone and make them happy. Like I said, I don’t want to bring another unhappy person into the world. I’ve been through such a hard time that I don’t want to repeat it.(M: INT. 13, 47, DO)

Ugh… always, for ten years one after another, one after another, one after another. He was always hitting me. He was always hitting me and talking to me badly. He has always… psychologically, and those things shouldn’t be done.(E: INT. 16, 37, DPsi)

Yes, but I have to have enough economic independence to take care of myself and do what I want at a certain moment, which I’m not doing right now. However, many of them confess to having personal reservations about committing to other partners as a result of previous intimate partner violence, except in cases where the partner has been or is a vital support (when they have not been the ones perpetrating the violence).(M: INT. 1, 32, DPsi)

Included? No, not at all, not at all. You can’t go around saying that you’re an abused woman, that you’ve been abused. It’s like… oh, trauma…, do you understand me?, and it… pushes you away.(M: INT. 5, 48, MU)

Yes, but I need to have enough financial independence to support myself and do what I want at a given moment, which I’m not currently doing.(M: INT. 1, 32, DPsi)

Thus, they indicate that psychological violence, accompanied by physical violence, is the general pattern of behavior exhibited by the perpetrators, who have also been a source of fear and insecurity to almost all victims, both during the violent situation and afterwards once they had managed to leave the situation. It is particularly significant that they link part of this violence to the psychological characteristics of the perpetrator [27,28]. Regardless of the diagnosis and age, the abusers are described as gamblers, substance abusers, aggressive, possessive, controlling, and jealous men.

He drank… I saw him drinking because downstairs there was his brother’s bar and every night he was stuck there and he didn’t come out, he didn’t come up to my house until eleven at night and when he went up, he went up really bombed… Our kids woke up when he hit me….(E: INT. 23, 29, DI)

Yes, sometimes I feel insecure. […] Because I have my idiot ex…(S: INT. 3, 25, DI)

But then he would insult me every day.(M: INT. 9, 39, DPsi)

Possessive, anxious, upset. Angry.(M: INT. 4, 29, DI)

Very jealous …He wouldn’t let me talk to my friends or anyone else.(E: INT. 23, 29, DI)

Fear. Fear of that person (perpetrator).(Y: INT. 22, 21, MU)

Yes, I was afraid of my husband, and towards the end, almost afraid of my last partner too.(Y: INT. 19, 57, PM)

To conclude this section, the discourse of the interviewed women with disabilities about the process of victimization reveals that the experiences mentioned have generated fear in all cases, whether it was in the past due to the timing of the violent episodes or still in the present, as a consequence of those incidents.

#### 4.2.2. Focus Group with Professionals

Among the conclusions drawn from the focus group, one key point is the need for public visibility of these instances of abuse, which often remain hidden for various reasons.

Firstly, there is a false assumption, held by a significant portion of society and the institutions responsible for their care, that these women are actually overprotected by their environment, that they are perpetually childlike, and that they do not experience any form of mistreatment. Their complaints are met with disbelief, especially in the case of women with intellectual disabilities. When they overcome their fear and dare to report it, their testimonies are often disregarded.

Furthermore, there is often insufficient attention paid to signs and indicators of gender-based violence. The social invisibility of these women creates collective blindness to warning signs that would typically be clear indicators for other victims.

You have said it, it is that we were made invisible, it is that a few years ago when talking about violence against women, they said no, no…, who is going to harm a woman with disability, who is going to harm a girl with disability?…(P2)

Even in Specialized Centers for Women Victims of Violence (Centros de Atención Especializada para Mujeres Víctimas de Violencia, CAVI), which provide assistance to women experiencing abuse, the support for women with disabilities is not considered one of their functions. There is also a lack of effective coordination between associations and regional or local administrations, with collaboration limited to certain phases of the process. Direct intervention and action primarily fall under the responsibility of the associations.

…I have to report…, okay, I have to report and then what…, which resources are there? … which resources do I have? I denounce so what, restraining order?; I have to leave my home?; I have no means, my situation is what it is… okay, but you know you can have support… it’s just that there aren’t steps.(P6)

According to experts’ opinions, the low number of reports from women with disabilities can be attributed to their vulnerable and high-risk situations. They may require assistance in going to police stations due to their type of disability, often accompanied by the very individuals who are abusing them. Fear, loneliness, social and personal isolation, lack of resources, and lack of support are insurmountable obstacles for them. Their dependency prevents them from making the decision to file a report.

Abusive caregivers who are caring for people with disabilities, with physical and organic disabilities, with serious mobility problems, and well, they do take care of them…(P4)

Additionally, they frequently end up normalizing the situation of violence, living with it, and accepting it as something ordinary. They are not aware that they are being abused. They perceive themselves as ‘damaged goods’, dependent on their abusers, and it is only when physical aggression occurs that they clearly recognize the mistreatment.

Lastly, it should be noted that the societal response to this violence is generally low. Society prefers not to acknowledge its existence or believes it only occurs in isolated cases. It is challenging to accept that a social system established in an advanced society and a welfare state, which should protect the most vulnerable groups, fails in its purpose. Discrediting testimonies and incapacitating those who dare to report it appears to be all too common. It is preferable to think of these cases as isolated incidents that occur in specific circumstances and environments, rather than acknowledging the magnitude of the problem as advocated by associations.

## 5. Discussion

Going a little deeper, it is interesting to compare the regional data obtained with the data at the national level (XIII Report of the State Observatory of Violence against Women and the Macro-Survey on Violence against Women 2019) [22,29], which coincide with the same categories of analysis used by the Ministry of Equality, IMSERSO, and INE.

In 2019, 9.1% of the victims of gender-based violence were women with disabilities, compared to 90.9% without recognized disabilities. From 2012 to 2019, the percentage of victims with disabilities was 9.8%. Therefore, being a woman with a disability means having almost a 1 in 10 probability of being a victim of gender-based violence resulting in death. In fact, the prevalence of violence in intimate relationships throughout life is higher among women with disabilities than among women without disabilities in all cases. For instance, 20.7% of women with recognized disabilities have experienced physical or sexual violence from a partner, compared to 13.8% of women without disabilities. In other countries, most studies report a significantly higher risk of victimization for women with disabilities than for persons without disabilities [30]; however, there are important regional differences. African women with disabilities are much more vulnerable to violence than western European or Asian women. Notably, 40.4% of women with disabilities have been abused by their partners, compared to 31.9% of women without disabilities (20.9% vs. 14.4% in the case of violence engendered by the current partner, and 52.2% vs. 42.9% in the case of violence occasioned by ex-partners). According to the Macro-Survey, 17.5% of women with disabilities who have experienced violence from a partner (9.8% from their current partner and 19.7% from previous partners) say that their disability is a consequence of the violence they have suffered. This percentage rises to 23.4% among women with disabilities who have experienced physical or sexual gender-based violence [22].

One of the findings from the regional study is that having a higher level of education or being a salaried worker is a protective factor against abuse (particularly general violence and physical abuse). However, it is important to note that in Spain, 2.5% of people with disabilities are illiterate, 21% have not completed secondary education, and 19% do not have a university degree [29]. Therefore, considering this variable as an inhibitor of violence is limited, as the majority have low educational levels and reduced rates of activity and employment [31], the international literature points out similar results [32]; thus, the empowerment of women with disabilities and their involvement in the Me Too movement has been limited to women with a higher economic and social status [33].

As shown in the narratives of the abused women (qualitative analysis), most of the women with disabilities in the regional study have managed to leave their partners. However, when comparing this information with the data from the Macro-Survey on Violence against Women [30], it is observed that the percentage of women who have left abusive partners is comparatively lower among women with recognized disabilities (78% compared to 69.2% for women without disabilities). Additionally, women with disabilities have experienced sexual violence to a greater extent (10.3%) than women without disabilities (6.2%). In other words, the number of women with disabilities who have suffered sexual abuse is higher (reaching 12% in the regional study), while the percentage of those who manage to leave their abusers is lower.

## 6. Conclusions

The conclusions drawn focus on both the detection of risk factors and possible protective measures against gender-based violence affecting women with disabilities.

The quantitative analysis concludes that the level of education acts as a protective factor. Women with disabilities who have a higher level of education (college or university degrees) are more able to defend themselves. These results align with national data, although it is also highlighted that this percentage, despite their education, is comparatively higher than that of women without disabilities [22]. In other words, uneducated women or women who received a non-higher level of education are more vulnerable, and they are even more so because of their disabilities.

Another variable used is the receipt of public economic assistance. Being a recipient of such assistance increases the risk of experiencing gender-based violence and being a victim of sexual abuse. These results indicate the perpetrators’ interest not only in controlling their partners’ economic resources but also in exerting power and control over them in other ways. Depriving them of economic resources impedes their autonomy and generates a situation of greater dependence and vulnerability.

Thirdly, membership in associative movements is a protective factor against physical abuse. Women who are part of associations have a public life where signs of abuse can be more easily noticed, and therefore, interventions can be made more quickly. Additionally, associative movements provide a space for assistance that can provide guidance to victims, resolve conflicts, and prevent abuse. The Macro-Survey on Violence against Women [22] indicates that 50.5% of women with disabilities who are victims of gender-based violence and 6.4% of those sexually harassed seek formal help from such services.

Lastly, a fourth variable that, after comparative analysis, emerges as a key protective factor against gender-based violence is employment status. Having paid work (regular or special, outside the home) reduces the chances of women with disabilities being victims of physical abuse, particularly sexual abuse. Spending time outside the home (which is conducive to intimate partner violence) and having economic capacity and independence reduces risks and promotes the social inclusion of these women. Unfortunately, the indicators of labor integration for this group are much lower than the rest of the population (an activity rate of 74.9% and an unemployment rate of 15.4% for the general population, compared to a 34.5% activity rate and 25.2% unemployment rate for people with disabilities [22]), and therefore, this protective barrier cannot be extended to a large portion of this population segment.

On the other hand, the fact that no significant values are found when crossing the rest of the variables (age, type, and degree of disability) with different forms of abuse (general, physical, sexual, or others) indicates that none of them is a determining risk or protective factor. Instead, what is truly important is the cross-cutting nature of gender-based violence.

The qualitative analysis yields the following conclusions. Firstly, self-esteem is a risk factor. The victims have a very negative perception of themselves. They even justify their abuse, especially if the aggressors are their partners, due to being disadvantaged and clumsy (‘damaged goods’), which makes them prone to continue suffering from that violence in current and future relationships.

These findings align with studies conducted in Spain on this subject [16]. In all situations of abuse, attacking a woman’s self-esteem is one of the first strategies of the aggressor to victimize their partners [34]. Therefore, it is crucial to act to improve their self-perception as an essential task of prevention and intervention. Therapeutic measures are necessary to address gender-based violence situations.

Other relevant aspects of the study include the descriptions and typologies of the aggressions and the profiles of the perpetrators. Acts of physical violence (pushing, hitting, kicking, and sexual abuse) are generally contextualized within the context of intimate partner relationships and are linked to specific psychological profiles of the abusers: jealous, controlling, violent individuals who abuse toxic substances.

Thirdly, there is a consensus that it is necessary to generate more support for women with disabilities. Testimonies from participants and experts conclude that it is crucial to promote support networks within the family and immediate environment (friends, reference centers, and work environments) as they provide essential support to address and resolve these situations. Otherwise, dependence on abusers and the normalization of violence condemn women to be permanent victims, putting their own lives at risk.

Another important issue is the visibility and social inclusion of women with disabilities. Ensuring that they become full-fledged citizens and are considered within the social structure to which they belong can reduce episodes of gender-based violence and, in the best case, make them disappear.

Furthermore, it is evident that women with disabilities who are victims of gender-based violence require specific and specialized support to address and prevent risky situations. Investing public funds in these actions has a direct impact on the quality of life of these women. It is essential to have properly trained professionals to provide assistance and counseling in the situations of fear and insecurity that arise after experiencing abuse. The implementation of specific plans and the effectiveness of institutional responses (coordinated and effective) are crucial.

Moreover, all information on this topic must be clear and accessible (including easy-to-read versions, among other things) so that women are aware of their rights and the available resources, thereby avoiding secondary victimization. This is something that the interviewees and experts themselves denounce.

Shifting focus, but continuing with proposals for action, it is important to emphasize the training and empowerment of women in terms of employment because economic independence can provide the possibility of escaping from episodes of violence or being able to overcome them. As demonstrated in this study, less than half of the participants earn their own income (from paid jobs, public assistance, or other sources), and only a quarter of them manage their income autonomously. Advancing in this direction implies increasing their capacity to control their own lives.

Finally, it is worth mentioning the limitations of this study. While the treatment of the collected data and the process of obtaining it have been rigorous, the power of these findings may lead to considering this research as an approximation or initial study of gender-based violence against women with disabilities at a regional level, without being able to make broader generalizations. However, it has been demonstrated that the sensitivity of the issue, coupled with the social and institutional tendency to minimize or overlook it, along with the limited presence, participation, and visibility of this collective in public life, hinders access to and, therefore, research on the subject. Nevertheless, none of this should prevent progress in this line of work as a necessary first step for subsequent intervention.

## Figures and Tables

**Figure 1 ijerph-20-06191-f001:**
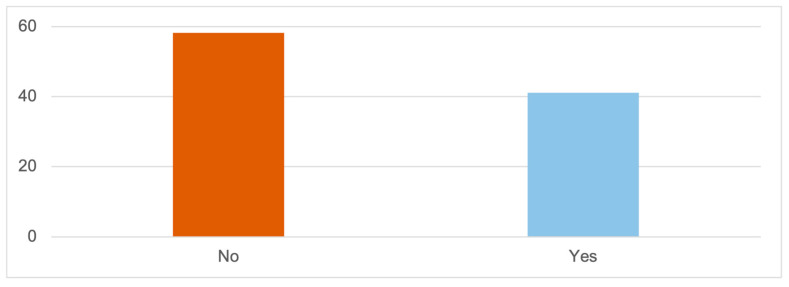
EXPERIENCED ABUSE (%); Source: own elaboration based on the data obtained during the survey.

**Figure 2 ijerph-20-06191-f002:**
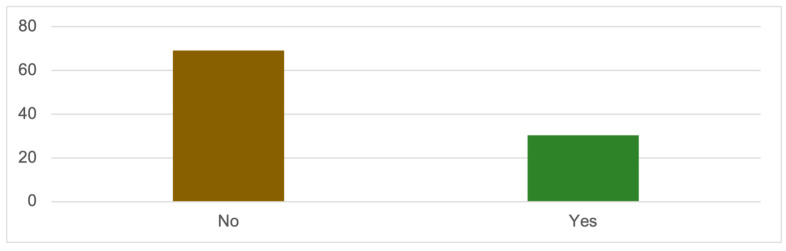
PHYSICAL ABUSE (%); Source: own elaboration based on the data obtained during the survey.

**Figure 3 ijerph-20-06191-f003:**
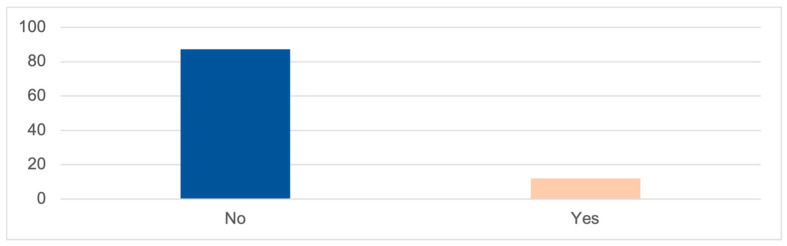
SEXUAL ABUSE (%); Source: own elaboration based on the data obtained during the survey.

**Figure 4 ijerph-20-06191-f004:**
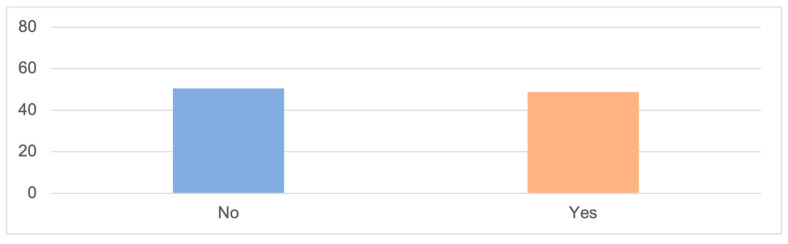
OTHER TYPES OF GENDER-BASED VIOLENCE (%); Source: own elaboration based on the data obtained during the survey.

**Table 1 ijerph-20-06191-t001:** Profile of organizations participating in the focus group.

Code	Organization Acronym	Organization Name	Profile of the Organization
P1	ONCE	Organización Nacional de Ciegos Españoles	For the autonomy and inclusion of blind people and people with visual disability
P2	CERMI	Comité Español de Representantes de Personas con Discapacidad	Platform for the representation, defense, and action of Spanish citizens with disabilities
P3	ACIME	Asociación Militares y Guardias Civiles con Discapacidad	Defense of the rights of soldiers and civil guards who have acquired a sudden disability
P4	FAMDIF	Federación de Asociaciones Murcianas de Personas con Discapacidad Física y Orgánica	Social and cultural inclusion of people with physical and organic disabilities for normalized active participation in society
P5	PI	Plena Inclusión	For the development of the quality-of-life project for people with intellectual or developmental disabilities and their families
P6	FUNDOWN	Fundación Síndrome de Down Región de Murcia	For the care and promotion of Personal Autonomy of the group of people with intellectual disabilities and/or Down Syndrome
P7	FUNDOWN		

Source: own elaboration based on the data obtained in the focus group.

**Table 2 ijerph-20-06191-t002:** Sections of the individual interview.

SECTION 1: Perception of Well-Being and Resilience		
Categories	Subcategories	
General life assessment		
		Life satisfaction
		Life purposes
		Self-esteem
Relational		
		Support network
		Sense of belonging and inclusion
Threats/resilience/emotional		
		Fear, insecurities
		Resilience/response strategies
		Professional emotional support
		General emotional state
SECTION 2: Perception of control and decision making		
Relationships and friendship		
		Current situation
		Motivations
		Limitations
Romantic		
		Relationships
		Sexual relationships
		Motivations
		Limitations
Health/reproductive		
		Health decisions and self-determination
		Access to healthcare
		Pregnancy and motherhood
Work/economic		
		Skills and motivations
		Access to belongings
		Economic independence
		Financial management
Independence/Control/Self-determination		
		Independence
		Control
		Self-determination
SECTION 3: Gender-based and/or intimate partner violence (sexual, physical, and psychological) and violence in close surroundings		
Home/close surroundings (non-partner)		
		Experience of violence
		Narrative/discourse of the experience
Partner		
		Experience of violence
		Narrative/discourse of the experience
Sexual		
		Sexual assault and harassment
		Narrative of the experience
Characteristics of abuse		
		Characteristics
		Frequency and temporality
Perpetrator		
		Relationship (past and present)
		Addictions
		Psychological profile (controlling, possessive, and abusive)
Victimhood 1		
		Experience
		Justification
		Support
Victimhood 2		
		Institutional response

Source: own elaboration based on the survey.

**Table 3 ijerph-20-06191-t003:** Sample of interviewed women.

CODE INT	C. Letter	CODE QUE	Age	Diagnoses	DIAG CODE	City
1	M	87	32	Psychosocial Disability	DPsi	Molina de Segura
2	M	102	74	Psychosocial Disability	DPsi	Cehegín
3	S	23	25	Intellectual Disability	DI	Cartagena
4	M	16	29	Intellectual Disability	DI	Cartagena
5	M	157	48	Multiple Diagnoses	Mu	Las Torres de Cotillas
7	Y	5	34	Intellectual Disability	DI	Murcia
8	N	88	53	Psychosocial Disability	DPsi	Molina de Segura
9	M	85	39	Psychosocial Disability	DPsi	Molina de Segura
10	C	105	66	Psychosocial Disability	DPsi	Mula
11	T	11	37	Mobility Problems	PM	Murcia
12	M	12	55	Physical Disability	DF	Murcia
13	M	154	47	Organic Disability	DO	Alguazas
14	P	54	49	Psychosocial Disability	DPsi	Cieza
15	M	52	37	Psychosocial Disability	DPsi	Cieza
16	E	44	37	Psychosocial Disability	DPsi	San Javier
19	Y	36	57	Mobility Problems	PM	San Pedro del Pinatar
22	Y	74	21	Multiple Diagnoses	Mu	Los Alcázares
23	E	76	29	Intellectual Disability	DI	San Javier

Source: own elaboration based on the data obtained during the survey.

**Table 4 ijerph-20-06191-t004:** Sociodemographic information.

Variable	N (% Valid)
Age	
Current age (*n* = 158)	Mean (SD): 44.21 (13.35)
	Median: 45.00
	Minimum: 21
	Maximum: 88
From 21 to 30 years	27 (17.1)
From 31 to 40 years	40 (25.3)
From 41 to 50 years	35 (22.2)
From 51 to 60 years	36 (22.8)
61 years and above	17 (10.8)
Place of residence	
Alone	22 (13.9)
Family members	116 (73.4)
Shared apartments	3 (1.9)
Residential facilities	7 (4.4)
Other options	10 (6.3)
Recognized degree of disability	
Recognized degree of disability (*n* = 158)	Median (SD): 65% (SD: 16.16)
33 to 64%	59 (37.3)
65 to 75%	53 (33.5)
Over 75%	19 (12.0)
Educational level	
Higher education	28 (17.7)
Non-higher education	127 (80.4)
Recognized disability	
Intellectual	91 (57.6)
Non-intellectual	67 (42.4)
Membership in the associative movement	
No	60 (38.0)
Unsure	17 (10.8)
Yes	81 (51.2)
Working status	
Employed	51 (32.3)
Unemployed	107 (67.7)
Financial assistance	
No	60 (38.0)
Unsure	21 (13.3)
Yes, myself	52 (32.9)
Yes, others	25 (15.8)

Source: own elaboration based on the data obtained during the survey.

**Table 5 ijerph-20-06191-t005:** Experienced abuse.

Variables	No		Yes		Chi	Df	sig
	*n*	%	*n*	%			
Educational level							
Higher education Non-higher education	21 69	75 54.8	**7** **57**	**25** **45.2**	**3.863**	1	0.049 *
Financial assistance							
No Unsure Yes, myself Yes, others	40 16 24 12	66.7 76.2 47.1 48.0	20 5 27 13	33.3 23.8 52.9 52	8.246	3	0.046 *

* Significance level: α = 0.05 (confidence interval: 95%); Source: own elaboration based on the data obtained during the survey.

**Table 6 ijerph-20-06191-t006:** Physical abuse.

Variables	No		Yes		Chi	Df	sig
	*n*	%	*n*	%			
Educational level							
Higher education Non-higher education	24 83	85.7 65.9	4 43	14.3 34.1	4.253	1	0.039 *
Membership in the associative movement							
No Unsure Yes	34 13 62	57.6 76.5 76.5	25 4 19	42.4 23.5 23.5	6.200	2	0.045 *
Working status							
Employed Unemployed	41 68	80.4 64.2	10 38	19.6 35.8	4.279	1	0.039 *

* Significance level: α = 0.05 (confidence interval: 95%); Source: own elaboration based on the data obtained during the survey.

**Table 7 ijerph-20-06191-t007:** Sexual abuse.

Variables	No		Yes		Chi	Df	sig
	*n*	%	*n*	%			
Working status							
Employed Unemployed	49 89	96.1 84.0	2 17	3.9 16.0	4.752	1	0.029 *

* Significance level: α = 0.05 (confidence interval: 95%); Source: own elaboration based on the data obtained during the survey.

## Data Availability

Data is unavailable due to privacy and ethical restrictions.

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
