# Peer review of "Health and Socioeconomic Determinants of Abuse among Women with Disabilities"

_ijerph, 2023, doi:10.3390/ijerph20126191_

Round 1

Reviewer 1 Report

The article presents a detailed analysis of a topic of current interest such as gender violence against women with disabilities. The objectives set are concrete, approachable, and relevant; the empirical methodology is adequate; and the results and discussion are consistent with the approach and the information collected. Finally, the conclusions are interesting, both academically and for professionals who care for women with gender violence.

Abstract: is informative and summarizes the work well; I suggest that the bibliographical references that appear be eliminated, leaving only the name of the scales.

Introduction: This section shows a good summary of the current conceptualization of disability. It adequately justifies the convenience of specifying gender violence against women with disabilities given the greater incidence on this group. I suggest that the notation of bibliographical references be reviewed, homogenizing the style and numbering in order of first appearance in the text.

Research Aims: correct, specific, and approachable with the information obtained.

Data and Methods: quantitative and qualitative methodology appropriate to the objectives set. The quantitative information obtained is correctly described. As in the previous section, the bibliographical references should be homogenized and renumbered.

Findings: the quantitative analysis is correct, analyzing the relationship between the sociodemographic variables and the forms of violence through the chi-square statistic for contingency tables. The qualitative analysis is adequate, obtaining results that allow deepening the effects of gender violence on the collective.

Discussion: an adequate comparison of the results obtained with those existing at the national level is made, although there is no reference to possible comparable results in other countries.

Conclusions: they are adequate, informative and supported by the information presented in the work.

References: I suggest that the references be reviewed, eliminating those that are not cited in the text and numbering them in order of appearance in the text.

Author Response

Dear Reviewer 1,

Thanks so much por your review and relevant and informative comments. Thanks to them,  the manuscript have been sustancial improved. Attached you will find  a summary table with the changes undergone by the authors. In this table, the different comments addressed have been mark with different colours (Green for reviewer 1, blue for reviewer 2 and purple for reviewer 3) so each one of the reviewer can easily trace which actions were taken and were in the manuscript the changes can be located.

Once again, we would love to show our gratitude for you valuable insight.

All the best,

The authors

Reviewer 2 Report

good article, please review the citations , they aren´t in the correct format, problems with the years, with parenthesis, bold,....and the form [XX] some of them are wrong. Review tables in one page instead of two.

examples:

pag 1, abstract: fogarty and brown (2002)

pag 3, apears citation 10, correctly, but not apear from 1 to 9 ....

pag 4: parentesis instead of square bracket in the citations

revise citation and bibliography

NOTES:

ABSTRACT

The research topic is clear, well structure and social interest.

The topic is threat in correct form and with a sufficient patients to compare

In general, the article is very interesting, clarifies the purpose of the study and I consider that the topic is in line with the journal’s research objectives.

INTRODUCTION:

The study objective is well defined and identified, the brief resume of tipes is well used.  In the introduction, well referenced but wrong cited.  Clear the aims of the article

The structure is well defined, and it has enough information in separated areas.

RESEARCH AINMS

The structure is well done, structured, clear and focused.

Different parts clearly defined and detailing.

DATA AND METHODS

The data are clear, well defined and explicated. The data are relevant.

The tables and graphics are clear, and focused on the topic of the article, better in one page.

The structured is well defined, and it has enough inofmation

DISCUSSION:

and the discussion are well drawn and interesting, well referenced. Well-structured in separated points.

An interesting line of research is observed, easy to go deep

Mentioned interesting line of another study for complementary this.

CONCLUSION: 

 An interesting line of research is observed, it could be go ahead in this line of research

Author Response

Dear Reviewer 2,

Thanks so much por your review and relevant and informative comments. Thanks to them,  the manuscript have been sustancial improved. Attached you will find  a summary table with the changes undergone by the authors. In this table, the different comments addressed have been mark with different colours (Green for reviewer 1, blue for reviewer 2 and purple for reviewer 3) so each one of the reviewer can easily trace which actions were taken and were in the manuscript the changes can be located.

Once again, we would love to show our gratitude for you valuable insight.

All the best,

The authors

Reviewer 3 Report

Overall feedback

1. The manuscript has merit in the form of the foundational arguments for the academic validity of the study.

2. When speaking about disability, perhaps the authors may want to interject the counter-argument that males also face gender based disability abuse and how this differs from the abuse faced by women, and how perhaps it is different for trans individuals. Then, the authors have to show relevant justification as to why the focus of this paper is on women and not men or trans individuals. Thank you.

Major issues

1. The authors have stated that they have used qualitative data for this research, but there is no mention in the methodology of the data analysis tool used to analyze the qualitative data. However, in the section on Findings the authors mention the use of content analysis. Please mention the use of content analysis in the methodology, and justify the use of content analysis versus other forms of qualitative data analysis (e.g. thematic analysis, coding). Please explain and address. 

2. The amount of qualitative data findings shared is somewhat small. May I please urge the authors to share more generously the qualitative findings from the abuse victims themselves since they are the focus of this study; it would be of great interest to the reader to hear their actual voices, and not just the voices of the focus group professionals. Additionally, in the conclusions the authors state that qualitative analysis yields certain conclusions, but the data for this is scant in the findings. Please may I request that the authors enrich their manuscript with more information from the qualitative data. 

Suggestions

1. The introduction is almost devoid of citation to support the assertions of the authors. May I please strongly urge the authors to amend this, thank you.

2. I find it strange that the first citation found in the text begins with the number 10 and then to the number 54. Where are the other citations? May I please advise the authors: if this manuscript is from a larger piece of work, i.e. a thesis, a chop/crop/paste approach is not the best option to take and the authors need to review the manuscript as an independent piece of work within its own self. Please address this by redoing the entire citations/references for this manuscript. Thank you. 

Author Response

Dear Reviewer 3,

Thanks so much por your review and relevant and informative comments. Thanks to them,  the manuscript have been sustancial improved. Attached you will find  a summary table with the changes undergone by the authors. In this table, the different comments addressed have been mark with different colours (Green for reviewer 1, blue for reviewer 2 and purple for reviewer 3) so each one of the reviewer can easily trace which actions were taken and were in the manuscript the changes can be located.

Once again, we would love to show our gratitude for you valuable insight.

All the best,

The authors

Round 2

Reviewer 3 Report

Thank you for addressing the concerns raised in the first round of peer review. The author(s) have made revisions that have uplifted the overall quality of the manuscript and have made the findings accessible and readable to the scholarly community.